## Research Article

Precision medicine; Pharmacogenomics; Psychotropics; Whole genome sequencing; Clinical implementation

**Corresponding author:**
Puthen Veettil Jithesh;
Email: jveettil@hbku.edu.qa

the Qatar Genome Program Research Consortium, A list of authors and their affiliations appears at the end of the paper.

# Clinically actionable pharmacogenomic landscape of antidepressants and antipsychotics in Qatar: a population-based cohort study

Dinesh Velayutham[1], Kholoud Bastaki[2], Areeba Irfan[1], Mohammed Abuhaliqa[3], Aisha AlMulla[1], the Qatar Genome Program Research Consortium, Suhaila Ghuloum[4], Muhammad Waqar Azeem[5], Munir Pirmohamed[6] and Puthen Veettil Jithesh[1,6] (ID)

[1]College of Health & Life Sciences, Hamad Bin Khalifa University, Doha, Qatar; [2]College of Pharmacy, QU Health, Qatar University, Doha, Qatar; [3]Research Department, Sidra Medicine, Doha, Qatar; [4]Department of Psychiatry, Hamad Medical Corporation, Doha, Qatar; [5]Department of Psychiatry, Sidra Medicine, Doha, Qatar and [6]Pharmacology & Therapeutics, Institute of Systems, Molecular and Integrative Biology, University of Liverpool, Liverpool, UK

## Abstract

Consortia like the Clinical Pharmacogenetic Implementation Consortium (CPIC) and the Dutch Pharmacogenetic Working Group (DPWG) provide clinical guidelines but pharmacogenomics implementation depends on population prevalence of actionable genetic variants and response phenotypes. We analyzed the distribution of actionable genetic variants and clinical recommendations in 14,354 adult Qataris, focusing only genes with guidelines (*CYP2C19*, *CYP2D6*, *CYP2B6* and *CYP3A4*). Haplotypes and diplotypes were generated from 490 alleles using whole genome data and metabolizer phenotypes were predicted based on current knowledge. Qatari population predicted to have actionable metabolizer phenotypes of CYP2C19, CYP2B6 and CYP2D6 impacting response to antidepressants were in the range of 1%–58% and for antipsychotics 0.1%–33% based on *CYP3A4* and *CYP2D6*. Fine-grained analysis based on clinical guidelines also revealed that while the Qataris may need prescription of an alternate antidepressant not metabolized by CYP2C19, patients from other populations may just need altering the dosage of tricyclic antidepressants like amitriptyline. Further studies incorporating other factors such as diet, environment and cultural habits alongwith population-specific variants will help in the pharmacogenomics implementation in the Qatari population.

## Impact statement

This study presents the largest pharmacogenomic landscape analysis of psychotropic drug metabolism in the Qatari population, examining the genes with guidelines for clinical implementation. Our findings reveal significant variability in metabolizer phenotypes, with up to 58% of individuals predicted to have altered metabolism for antidepressants and up to 33% for antipsychotics. Population-specific differences were observed. For example, the major recommendation for the Qatari population based on the actionable frequencies would be to prescribe alternative antidepressants that are not metabolized by CYP2C19 such as those used for tricyclic antidepressants (TCAs), while individuals from other populations may only require dosage adjustments of the same TCAs. The study provides a foundation for the potential clinical integration of pharmacogenomics in psychotropic medication management in Qatar. However, further studies are required on genotype-to-phenotype translation, including the contribution of population-specific variants and inclusion of environmental factors while predicting response.

## Introduction

Antidepressants and antipsychotics are widely prescribed in Qatar and worldwide but response (efficacy and safety) can be highly variable (Bastaki et al., 2021a, 2021b). Genetic variants contribute to this inter-individual variability and pharmacogenetic testing provides a means to predict potential non-response or adverse response (Bousman et al., 2023a; Pirmohamed, 2023). Guidelines for the clinical implementation of pharmacogenetic tests for several psychotropic medications have been provided by the Clinical Pharmacogenetic Implementation Consortium (CPIC) (Bousman et al., 2023b; Hicks et al., 2017) and the Dutch Pharmacogenetic Working Group (DPWG) (Beunk et al., 2024). Based on these guidelines, it is possible to define clinically

actionable genotypes or diplotypes (for highly polymorphic genes) and their predicted phenotypic consequences that affect response to specific drugs. Understanding the distribution of clinically actionable genotypes/diplotypes and their predicted response phenotypes is essential for facilitating the clinical implementation of pharmacogenomics (PGx) in various healthcare settings (Jithesh and Scaria, 2017).

Genome studies of populations under-represented in the large genomic datasets are gaining momentum (Sirugo et al., 2019). The Qatar Genome Program (QGP) has provided such an opportunity to study not only the native Qatari population but also extend the findings to several other populations in the Middle East region as well (Mbarek et al., 2022). We previously identified a differential distribution of actionable frequencies across various gene-drug combinations among Qataris and other world populations using the QGP pilot phase data (Jithesh et al., 2022). In this study, we focused on clinical guidelines available for results from PGx tests predicting the metabolizer status of the cytochrome P450 enzymes, CYP2C19, CYP2D6, CYP2B6 and CYP3A4, which are known to affect the pharmacokinetics of several antidepressants and antipsychotics. Fine-grained analysis was conducted on a dataset of more than 14,000 whole genomes from the population to reveal the differential distribution of such recommendations across populations compared to other world populations from the thousand genomes dataset.

## Methods

Following approval from the Institutional Review Board, we studied an observational cohort of 14,699 adult Qataris recruited by the Qatar Biobank (QBB) between 11 December 2012 and 9 June 2016. Their genomes were sequenced as part of the Qatar Genome Program (QGP) and 14,354 were taken forward following quality analysis of the whole genome sequencing data. Further details of the pilot cohort and genome data processing have been presented previously (Jithesh et al., 2022).

We analyzed genes coding for the metabolizing enzymes, CYP2C19, CYP2D6, CYP2B6 and CYP3A4, that significantly affect response to psychotropics and also have guidelines for clinical interpretation from either CPIC or DPWG (Beunk et al., 2024; Bousman et al., 2023b; Hicks et al., 2017). We extracted 490 alleles to generate haplotypes and diplotypes and predicted the associated metabolizer phenotypes based on PharmVar (https://www.pharmvar.org/) and CPIC translation tables (https://cpicpgx.org/genes-drugs/). Actionability was defined according to clinical guidelines from CPIC and DPWG (evidence Level 1A PharmGKB), which recommend specific actions such as dosage adjustments or alternative drug prescriptions for individuals with certain diplotypes and their predicted metabolizer phenotypes for the above genes. While other genes such as *ABCB1*, *SLC6A4* and *HTR2A* are known to be associated with psychotropic response, they do not meet Level 1A evidence criteria for clinical actionability and hence were not included in our study. We used all the alleles that are known to contribute towards the metabolizer phenotype from PharmVar, including SNVs and structural variants. To facilitate this, we used BAM files to call PGx star alleles using Aldy (Hari et al., 2023). Cyrius (https://github.com/Illumina/Cyrius) was used for CYP2D6 star allele calling recalculating the activity scores for CYP2D6 to consider the complex/unique copy number changes in the QGP data. We observed 298 combinations of *CYP2D6* alleles in our

data set and Supplementary Figure 1 illustrates their predicted metabolizer status. For statistical presentation of results, we used absolute numbers and percentages. To compare proportions between two populations, we employed a two-proportions z-test, calculating confidence intervals to quantify the uncertainty around these differences. Additionally, we computed odds ratios as a measure of effect size to assess the magnitude of differences for change in recommendation proportions. The Benjamini-Hochberg method was applied for multiple testing correction and significance was evaluated at a genome-wide significance threshold of $5 \times 10^{-8}$ and additional correction was performed for the 13 drugs tested. For serotonin reuptake inhibitors, the CPIC guidelines for *CYP2C19*, *CYP2B6* and *CYP2D6* were used for calculating actionable frequencies. For tricyclic antidepressants (TCAs), individual and combined *CYP2C19* and *CYP2D6* guidelines from CPIC were used. For antipsychotics, *CYP2D6* and *CYP3A4* guidelines from DPWG were used. We also utilized the psychotropic medication prescription pattern from the Qatari Mental Health Hospital in our previous studies (Bastaki et al., 2021a, 2021b) to infer the potential implications of our findings from the genome data on healthcare in Qatar.

## Results

Genome sequencing data from 14,354 Qataris revealed the distribution of the actionable diplotypes and associated phenotypes potentially affecting the response to several antidepressants and antipsychotics in this population (Table 1).

### Serotonin reuptake inhibitors and serotonin modulators

For the serotonin reuptake inhibitors, including selective serotonin reuptake inhibitors (SSRIs), serotonin and norepinephrine reuptake inhibitors (SNRIs) and SSRI-like serotonin modulators, the frequency of clinically actionable diplotypes and their predicted phenotypes ranged from just over 1% to close to 59%. Clinical guidelines are available from the CPIC based on CYP2C19 metabolizer status for the SSRIs citalopram and escitalopram (Supplementary Table 1). Citalopram/escitalopram had the highest actionable frequencies of all the psychotropics evaluated (n = 8,420; 58.7%) based on CYP2C19 ultrarapid, rapid, poor or intermediate metabolizer status. Clinical guidelines for some other SSRIs such as fluvoxamine and paroxetine as well as venlafaxine (an SNRI) and vortioxetine (a serotonin modulator) are based on the diplotypes and metabolizer status of CYP2D6. For fluvoxamine and venlafaxine only CYP2D6 poor metabolizers were considered to be actionable and hence had the lowest frequency (1.2%) among the antidepressants studied. Five SSRIs, escitalopram, fluoxetine, paroxetine, sertraline and fluvoxamine are prescribed in the Qatari population. We calculated the actionable percentage for all SSRIs except fluoxetine. Escitalopram accounts for 26% of all the antidepressant prescriptions in Qatar (Bastaki et al., 2021b), with 58% of those predicted to carry actionable variants affecting its metabolism as per this study.

When the distribution of the actionable phenotypes in the Qatari population was compared with other world populations represented in the 1,000 genomes project, some significant differences were observed (Table 1). Actionable frequencies were significantly different in the case of fluvoxamine (adjusted p-val: $3.8 \times 10^{-5}$) and paroxetine (adj p = $1 \times 10^{-5}$), though no significant

**Table 1.** The distribution of actionable phenotypes predicted from diplotypes affecting response to psychotropics and requiring alteration of dosage or alternate prescription in the Qatari population based on 14,354 whole genomes

| Class of drug | Drugs (Pharmacogenes) | Metabolizer phenotype | QGP # (%) | 1KG # (%) | 1KG_EU # (%) | Total actionable # (%) | | |
| --- | --- | --- | --- | --- | --- | --- | --- | --- |
| | | | | | | QGP | 1KG | 1KG_EU |
| Antidepressants – selective serotonin reuptake inhibitors (SSRIs)[a] | Citalopram, Escitalopram (CYP2C19) | UM | 959 (6.7%) | 101 (3.15%) | 26 (4.11%) | 8,420 (58.7%) | 1712 (53.47%) | 299 (47.31%) |
| | | PM | 275 (1.9%) | 205 (6.4%) | 7 (1.11%) | | | |
| | | RM | 4,184 (29.1%) | 328 (10.2%) | 99 (15.66%) | | | |
| | | IM | 2,997 (20.9%) | 1,078 (33.67%) | 167 (26.42%) | | | |
| | Sertraline (CYP2C19 + CYP2B6) | C19 UM/RM + B6 UM/RM | 105 (0.7%) | 11 (0.34%) | 3 (0.47%) | 6,030 (42.01%) | 1,627 (50.87%) | 203 (32.12%) |
| | | C19 IM + B6 NM | 1,308 (9.1%) | 369 (11.5%) | 62 (9.81%) | | | |
| | | C19 NM + B6 IM | 2,170 (15.1%) | 219 (6.83%) | 35 (5.54%) | | | |
| | | C19 IND + B6 IM | 16 (0.1%) | 70 (2.18%) | 11 (1.74%) | | | |
| | | C19 IM + B6 IND | 213 (1.5%) | 213 (6.65%) | 32 (5.06%) | | | |
| | | C19 IM + B6 IM | 1,136 (7.9%) | 203 (6.3%) | 21 (3.32%) | | | |
| | | C19 PM + B6 NM | 117 (0.8%) | 63 (1.9%) | 1 (0.16%) | | | |
| | | C19 PM + B6 IM | 107 (0.74%) | 48 (1.49%) | 1 (0.16%) | | | |
| | | C19 PM + B6 IND | 21 (0.15%) | 42 (1.3%) | 2 (0.32%) | | | |
| | | C19 PM + B6 UM/RM | 7 (0.05%) | 1 (0.03%) | 0 | | | |
| | | C19 NM + B6 PM | 527 (3.67%) | 160 (5%) | 18 (2.85%) | | | |
| | | C19 IND + B6 PM | 10 (0.07%) | 55 (1.72%) | 5 (0.79%) | | | |
| | | C19 IM + B6 PM | 265 (1.9%) | 150 (4.68%) | 12 (1.90%) | | | |
| | | C19 PM + B6 PM | 23 (0.2%) | 25 (0.78%) | 0 | | | |
| | Fluvoxamine (CYP2D6) | PM | 166 (1.2%) | 95 (3%) | 44 (6.96%) | 166 (1.2%) | 95 (3%) | 44 (6.96%) |
| | Paroxetine (CYP2D6) | UM | 1,230 (8.6%) | 107 (3.3%) | 20 (3.16%) | 4,729 (32.9%) | 1,286 (40.16%) | 304 (48.10%) |
| | | PM | 166 (1.2%) | 95 (3%) | 44 (6.96%) | | | |
| | | IM | 3,331 (23.2%) | 1,084 (33.9%) | 240 (37.97%) | | | |
| Antidepressants – serotonin & norepinephrine reuptake inhibitors SNRIs[a] | Venlafaxine (CYP2D6) | PM | 166 (1.2%) | 95 (3%) | 44 (6.96%) | 166 (1.2%) | 95 (3%) | 44 (6.9%) |
| Antidepressants – SSRI-like serotonin modulators[a] | Vortioxetine (CYP2D6) | UM | 1,230 (8.6%) | 107 (3.34%) | 20 (3.16%) | 1,396 (9.7%) | 202 (6.3%) | 64 (10.13%) |
| | | PM | 166 (1.2%) | 95 (3%) | 44 (6.96%) | | | |
| Antidepressants – TCAs[v] | Amitriptyline clomipramine doxepin imipramine trimipramine (CYP2C19 + CYP2D6) | C19 UM/RM + D6 UM/RM | 484 (3.4%) | 19 (0.6%) | 2 (0.32%) | 7,991 (55.7%) | 1,468 (44.9%) | 313 (49.52%) |
| | | C19 UM/RM + D6 PM | 52 (0.4%) | 15 (0.5%) | 7 (1.11%) | | | |
| | | C19 IM + D6 UM/RM | 198 (1.4%) | 28 (0.9%) | 8 (1.27%) | | | |
| | | C19 PM + D6 UM/RM | 17 (0.1%) | 3 (0.09%) | 0 | | | |
| | | C19 PM + D6 IM | 63 (0.4%) | 63 (2%) | 2 (0.32%) | | | |
| | | C19 PM + D6 PM | 5 (0.03%) | 3 (0.09%) | 1 (0.16%) | | | |
| | | C19 NM + D6 UM/RM | 530 (3.7%) | 39 (1.2%) | 7 (1.11%) | | | |
| | | C19 UM/RM + D6 NM | 3,103 (21.6%) | 222 (7%) | 54 (8.54%) | | | |

**Table 1.** (*Continued*)

| Class of drug | Drugs (Pharmacogenes) | Metabolizer phenotype | QGP # (%) | 1KG # (%) | 1KG_EU # (%) | Total actionable # (%) | | |
|---|---|---|---|---|---|---|---|---|
| | | | | | | QGP | 1KG | 1KG_EU |
| | | C19 UM/RM + D6 IM | 1,153 (8.04%) | 148 (4.6%) | 56 (8.86%) | | | |
| | | C19 PM + D6 NM | 165 (1.14%) | 118 (3.7%) | 3 (0.47%) | | | |
| | | C19 NM + D6 PM | 77 (0.53%) | 34 (1.1%) | 19 (3.01%) | | | |
| | | C19 IM + D6 PM | 32 (0.2%) | 28 (0.9%) | 10 (1.58%) | | | |
| | | C19 NM + D6 IM | 1,331 (9.3%) | 356 (11.1%) | 92 (14.56%) | | | |
| | | C19 IM + D6 IM | 779 (5.4%) | 392 (12.2%) | 52 (8.23%) | | | |
| Antipsychotics¥ | Quetiapine (CYP3A4) | PM | 14 (0.1%) | 1 (0.0003%) | 1 (0.16%) | 14 (0.1%) | 1 (0.0003%) | 1 (0.15%) |
| | Aripiprazole Brexpiprazole (CYP2D6) | PM | 166 (1.2%) | 95 (3%) | 44 (6.96%) | 166 (1.2%) | 95 (3%) | 44 (6.9%) |
| | Haloperidol Risperidone (CYP2D6) | UM | 1,230 (8.6%) | 107 (3.34%) | 20 (3.16%) | 1,396 (9.7%) | 202 (6.3%) | 64 (10.13%) |
| | | PM | 166 (1.2%) | 95 (3%) | 44 (6.96%) | | | |
| | Pimozide (CYP2D6) | PM | 166 (1.2%) | 95 (3%) | 44 (6.96%) | 3,499 (24.3%) | 1,179 (36.8%) | 284 (44.94%) |
| | | IM | 3,331 (23.2%) | 1,084 (33.9%) | 240 (37.97%) | | | |
| | Zuclopenthixol (CYP2D6) | PM | 166 (1.2%) | 95 (3%) | 44 (6.96%) | 4,729 (32.9%) | 1,286 (40.2%) | 304 (48.10%) |
| | | IM | 3,331 (23.2%) | 1,084 (33.9%) | 240 (37.97%) | | | |
| | | UM | 1,230 (8.6%) | 107 (3.34%) | 20 (3.16%) | | | |

QGP: Qatar Genome Program; 1KG: Thousand genomes; 1KG_EU: European superpopulation data from the 1,000 genomes.
Metabolizer status – UM: ultrarapid; RM: rapid; PM: poor; IM: intermediate; NM: normal; IND: indeterminate.[a]For serotonin reuptake inhibitors, the CPIC guidelines[5] for CYP2C19 (C19), CYP2B6 (B6) and CYP2D6 (D6) were used for calculating actionable frequencies. ¥For tricyclic antidepressants (TCAs), individual and combined C19 and D6 guidelines from CPIC[4] were used. ¥ For antipsychotics, D6 and CYP3A4 guidelines from DPWG[6] were used.

difference was observed for citalopram/escitalopram. Further differences were identified when compared to the European population present in the 1,000 genomes project as well, with lower actionable frequencies in the Qataris compared to Europeans for both fluvoxamine and paroxetine (Table 2).

In the case of sertraline, guidelines for clinical action are based on both CYP2C19 and CYP2B6 metabolizer status and hence a number of combinations are considered actionable (Table 1). Based on these combinations, 42% of the population may have to alter the dosage of sertraline or be prescribed an alternate antidepressant. This is significantly lower compared to other world populations in the thousand genomes (51%; adj p = $1.7 \times 10^{-10}$) but higher compared to the Europeans (32%) though this did not reach statistical significance (adj p > 0.05) (Table 2).

We further analyzed the data to see whether there were differences in the distribution of specific clinical implementation recommendations for sertraline. Most individuals across all populations fell into the category for which the recommendations is to initiate therapy with the standard starting dose and to consider a slower titration schedule and a lower maintenance dose (Figure 1). However, additional categories of recommendations showed differences in distribution between the Qatari population and the other populations in the 1,000 genomes dataset. Notably, for the categories of recommendations suggested to alter the starting dose or replacement of sertraline with another antidepressant, the Qatari population showed a lower frequency compared to the 1,000 genomes populations. On the contrary, for the categories where suggestions were to initiate treatment with the recommended starting dose and to make alterations if required, the

Qatari population had a higher frequency compared to other world populations (Figure 1).

### Tricyclic antidepressants (TCAs)

For amitriptyline, we considered the CPIC clinical guidelines based on the combination of *CYP2C19* and *CYP2D6* diplotypes for calculating actionability: approximately 56% of the individuals studied may have an actionable TCA metabolizer phenotype associated with diplotypes in these genes which could potentially affect response. The same guideline can be used for other TCAs such as clomipramine, doxepin, imipramine and trimipramine according to CPIC and hence would require modification to the prescription in more than half of the Qatari population based on the diplotypes of the two genes (Table 1). TCAs, including amitriptyline (alone and in combination with chlordiazepoxide), nortriptyline with fluphenazine, clomipramine, imipramine and trimipramine are prescribed to the Qatari population (Bastaki et al., 2021b). We calculated the actionable counts individually for amitriptyline, clomipramine, imipramine and trimipramine, excluding nortriptyline + fluphenazine and chlordiazepoxide. Among all antidepressant prescriptions, amitriptyline accounts for 23% of cases. Based on genetic predictions, more than half of the Qatari population carries actionable variants affecting TCA metabolism, which translates to approximately 12% of mental health patients being at risk of altered drug response potentially leading to inefficacy or adverse drug reactions.

Further, investigating the differences in distribution of specific clinical implementation recommendations for TCAs, we found

**Table 2.** Comparison of actionable proportions from the Qatari population (QGP) with other world populations present in the 1,000 genomes dataset. P-values from the two-proportions z-test are provided for each drug

| | Medications | QGP | 1KG | 1KG_EU |
|---|---|---|---|---|
| Antidepressants – selective serotonin reuptake inhibitors (SSRIs) | Citalopram | 58.7 | 53.5 | 47.31 |
| | Sertaline ** | 42.01 | 50.87 | 32.12 |
| | Fluvoxamine * | 1.2 | 3 | 6.96 |
| | Paroxetine * Ω | 32.9 | 40.16 | 48.1 |
| Antidepressants – serotonin & norepinephrine reuptake inhibitors SNRIs | Veniafaxine * ΩΩ | 1.2 | 3 | 6.96 |
| Antidepressants – SSRI-like serotonin modulators | Vortioxetine | 9.7 | 6.3 | 10.13 |
| Antidepressants – TCAs | Amitriptyline ++ ** | 55.7 | 44.9 | 49.52 |
| Antipsychotics | Quetiapine | 0.1 | 0.0003 | 0.15 |
| | Aripiprazole * ΩΩ | 1.2 | 3 | 6.9 |
| | Brexpiprazole * ΩΩ | 1.2 | 3 | 6.9 |
| | Risperidone + Haloperidol | 9.7 | 6.3 | 10.13 |
| | Pimozide ** ΩΩ | 24.3 | 36.8 | 44.94 |
| | Zuclopenthixol * Ω | 32.9 | 40.2 | 48.1 |

*P values from QGP vs. 1KG.
Ω P values from QGP vs. 1KG-EU.
*<0.5E−05 and **<0.5E−08. Ω <0.5E−05 and ΩΩ < 0.5E−08. Two-way Z-test for proportionates between QGP, 1KG and 1KG-EU after correcting for $10^{-8}$ and the 13 drugs tested.

that the major recommendation, based on the diplotypes of *CYP2C19* and *CYP2D6*, for the Qatari population was to consider an alternative drug not metabolized by CYP2C19 (Figure 2). However, in the case of populations represented in the thousand genomes dataset, the major suggestion was to consider a 25% reduction of the recommended starting dose of amitriptyline and other TCAs.

### Antipsychotics

For several antipsychotics, DPWG guidelines allow clinical actionability based on the diplotypes of *CYP2D6*. For instance, CYP2D6 poor metabolizers are alone considered actionable for aripiprazole and brexpiprazole and were observed in 1.2% of the population. In the case of haloperidol and risperidone, both poor and ultrarapid metabolizers are clinically actionable (9.7% in the Qataris). Pimozide prescriptions need alteration based on poor and intermediate metabolizer status (24.3%) while for zuclopenthixol, in addition to the above two ultrarapid metabolizers need action (~33%). Atypical antipsychotics including quetiapine and risperidone are commonly prescribed in the Qatari population (Bastaki et al., 2021a). Quetiapine accounts for 16% of all antipsychotic prescriptions with 0.10% of the population predicted to carry actionable variants affecting its metabolism by this analysis. Similarly, risperidone is prescribed in 13% of cases with 9.7% of the population carrying actionable variants that may influence drug response (Supplementary Table 3).

DPWG also provides guidelines for the use of *CYP3A4* diplotypes and the associated metabolizer phenotypes for predicting

the risk of non-response to quetiapine. However, only CYP3A4 poor metabolizers need to be considered for the appropriate action and the number of participants with this metabolizer status as predicted from their diplotypes was quite low in the population (n = 14; 0.1%) (Supplementary Table 2). Quetiapine, aripiprazole, brexpiprazole, pimozide and zuclopenthixol had a significantly lower proportion of actionable variants in the QGP when compared with the European population (Table 1).

### Discussion

We predicted the distribution of CYP2B6, CYP2C19, CYP2D6 and CYP3A4 metabolizer phenotypes in 14,354 Qatari individuals, incorporating multiple variants within each gene, including copy number variations. Based on the latest PGx guidelines for antidepressant and antipsychotic prescribing, we found that the distribution of actionable genotypes in the Qatari population differs significantly from other global populations, such as those represented in the 1,000 genomes dataset. These differences underscore the importance of studying underrepresented populations, as reliance on global datasets may not accurately capture the unique genetic architecture of distinct populations. For example, more than half of the Qatari population may require alterations in escitalopram and amitriptyline prescriptions based on CPIC recommendations. Given that both these drugs are among the most commonly prescribed antidepressants in Qatar (Bastaki et al., 2021b), these findings emphasize the potential impact of PGx testing in optimizing medication efficacy and safety.

Further fine-grained analysis revealed population-specific differences in PGx-based clinical recommendations. Notably, while Qataris with certain *CYP2C19* genotypes may benefit from alternative antidepressants not metabolized by this enzyme, individuals from other populations may only require dose adjustments for tricyclic antidepressants such as amitriptyline. The successful integration of PGx into routine clinical care will require addressing real-world challenges, including infrastructure development, clinician education and patient access to testing (Jarvis et al., 2023). As outlined in our clinical implementation study on the Qatari population, such efforts will be critical for developing formulary guidelines and precision medicine strategies tailored to the region (Bastaki et al., 2024).

A limitation of this study is that phenotype predictions were based on existing literature from other populations. While it is generally assumed that the functional effects of diplotypes remain consistent across populations, population-specific genetic architecture, linkage disequilibrium patterns and environmental factors like diet may influence the phenotype expression. Studies incorporating functional characterization of the population-specific variants and real-world clinical response outcomes in the Qatari population will be essential to confirm these predictions. We are actively developing machine learning algorithms capable of predicting the functional impact of novel and rare variants in pharmacokinetic genes and these models will integrate multi-omics data, structural predictions and functional annotations to improve the characterization of rare alleles. It is worth noting that the Qatari population data we studied includes several subpopulations, which are present in the Middle Eastern region at varying proportions. Thus, we presume extending the results from Qatar to other Middle Eastern populations will not lead to substantial inaccuracies.

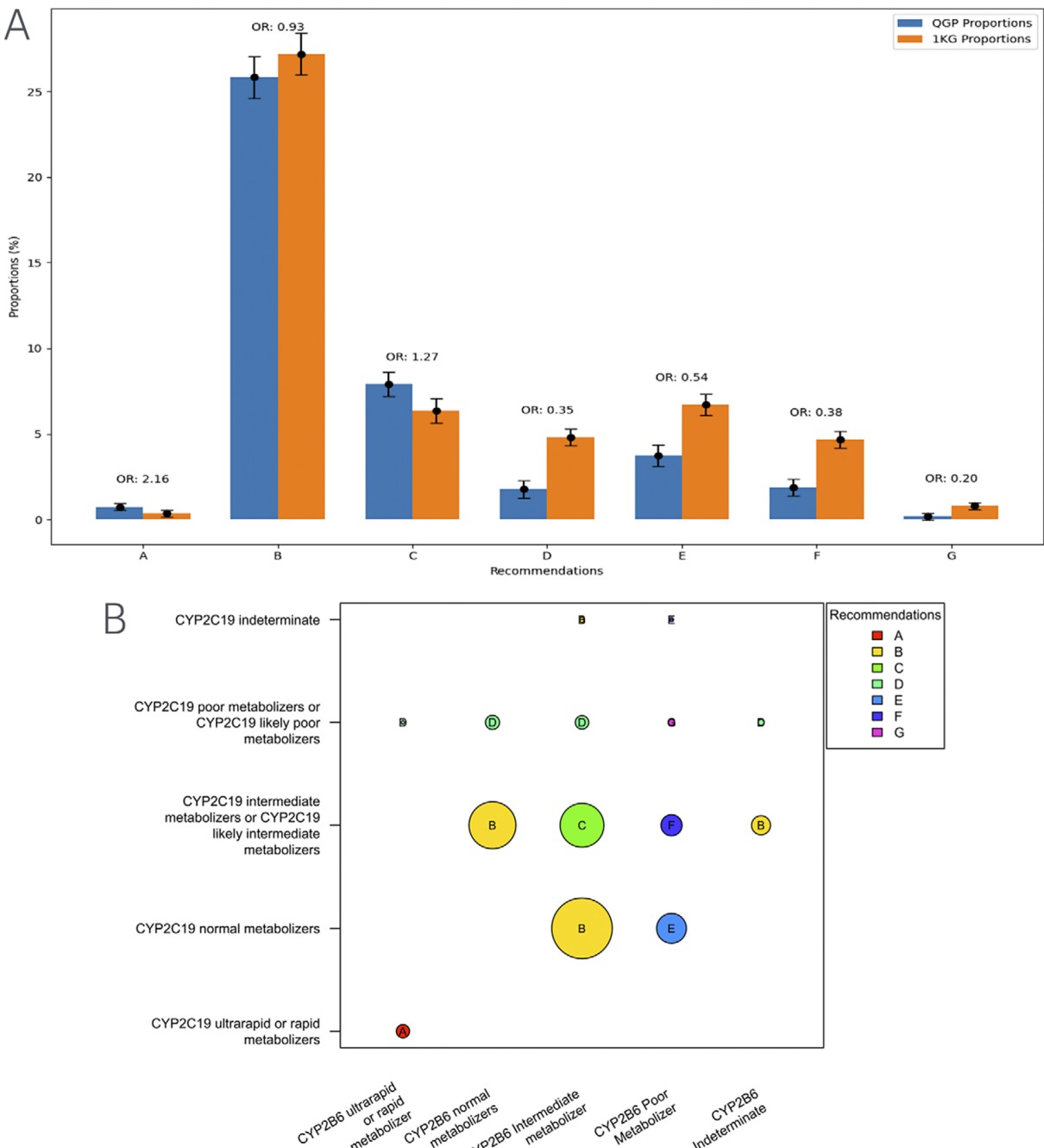

**Figure 1.** (A) Distribution of clinical implementation recommendations for sertraline based on the combined CYP2C19-CYP2B6 metabolizer status in the Qatari population and the 1,000 genomes. (B) Visualization of the computation of recommendations for the combined metabolizer status of CYP2C19 and CYP2B6. Various categories from A–G in the figure are as below.

(A) Distribution of clinical implementation recommendations for sertraline based on the combined CYP2C19-CYP2B6 metabolizer status in the Qatari population and the 1,000 genomes. (B) Visualization of the computation of recommendations for the combined metabolizer status of CYP2C19 and CYP2B6. Various categories from A–G in the figure are as below.

A Initiate therapy with recommended starting dose. If patient does not adequately respond to recommended maintenance dosing, consider titrating to a higher maintenance dose or switching to a clinically appropriate alternative antidepressant not predominantly metabolized by CYP2C19 or CYP2B6

B Initiate therapy with recommended starting dose. Consider a slower titration schedule and lower maintenance dose.

C Initiate therapy with recommended starting dose. Consider a slower titration schedule and lower maintenance dose than normal metabolizers.

D Consider a lower starting dose, slower titration schedule and 50% reduction of standard maintenance dose as compared to CYP2C19 normal metabolizers or select a clinically appropriate alternative antidepressant not predominantly metabolized by CYP2C19.

E Consider a lower starting dose, slower titration schedule and 25% reduction of standard maintenance dose as compared to CYP2B6 normal metabolizers or select a clinically appropriate alternative antidepressant not predominantly metabolized by CYP2B6.

F Consider a lower starting dose, slower titration schedule and 50% reduction of standard maintenance dose as compared to CYP2B6 normal metabolizers.

G Select an alternative antidepressant not primarily metabolized by CYP2C19 or CYP2B6.

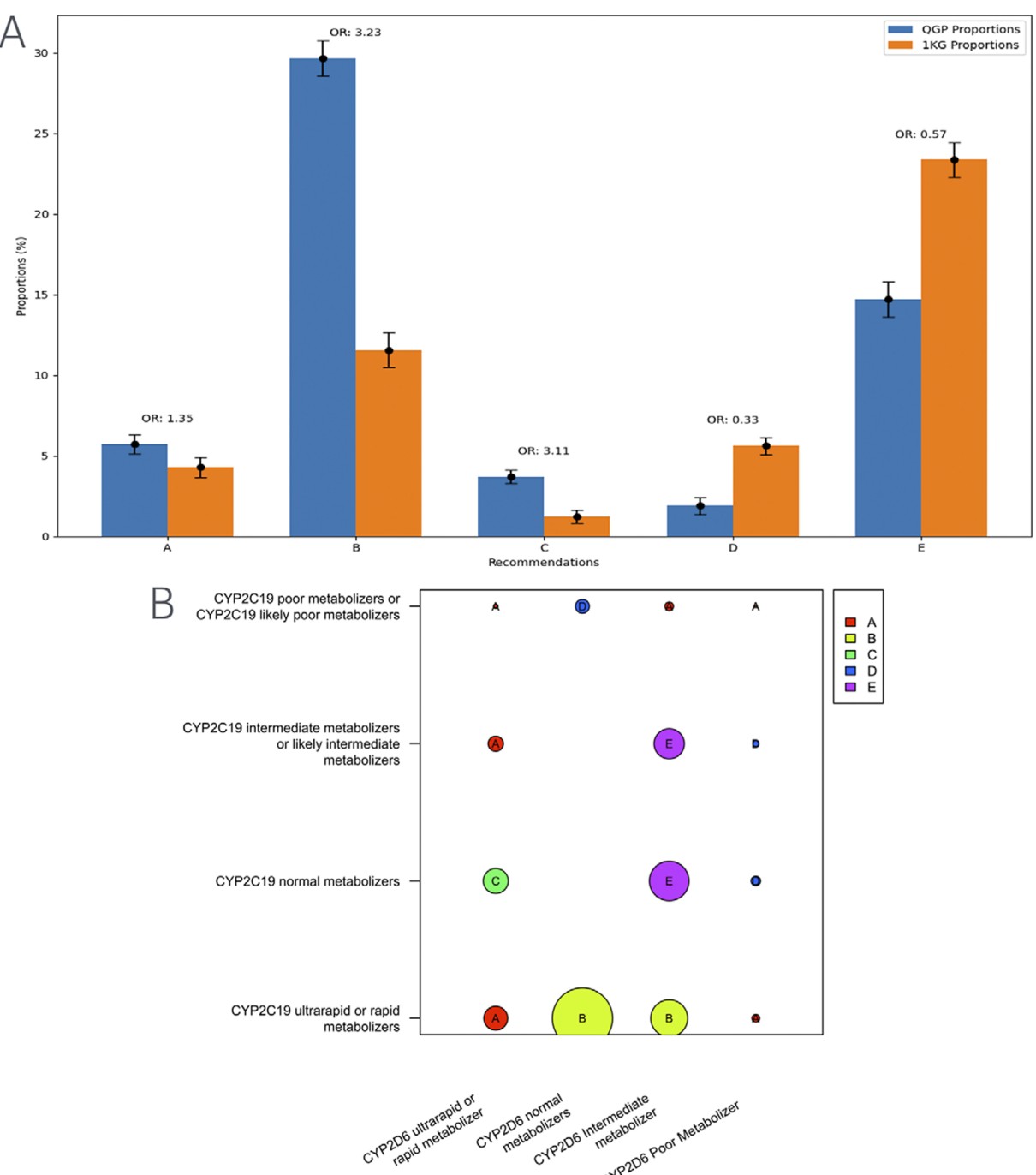

**Figure 2.** Distribution of clinical implementation recommendations for tricyclic antidepressants such as amitriptyline based on the combined CYP2C19-CYP2D6 metabolizer status in the Qatari population and the 1,000 genomes. (B) Visualization of the computation of recommendations for the combined metabolizer status of CYP2C19 and CYP2D6. Various categories from A–E in the figure are as below.
A Avoid amitriptyline use
B Consider alternative drug not metabolized by CYP2C19.
C Avoid amitriptyline use. If amitriptyline is warranted, consider titrating to a higher target dose (compared to normal metabolizers).
D Avoid amitriptyline use. If amitriptyline is warranted, consider a 50% reduction of recommended starting dose.
E Consider a 25% reduction of recommended starting dose.

In conclusion, this study highlights the high prevalence of clinically actionable pharmacokinetic gene variants affecting the metabolism of commonly prescribed psychotropic medications in the Qatari population. Based on internationally accepted PGx guidelines, a substantial proportion of patients in Qatar may be predicted to require drug dose modifications or alternative therapy selection to enhance treatment efficacy and safety. These findings provide a strong rationale for implementing PGx testing in clinical practice, ultimately paving the way for a more personalized approach to psychiatric medication management in Qatar and beyond.

**Open peer review.** To view the open peer review materials for this article, please visit http://doi.org/10.1017/pcm.2025.2.

**Supplementary material.** The supplementary material for this article can be found at http://doi.org/10.1017/pcm.2025.2.

**Data availability statement.** The informed consent given by the study participants does not cover the posting of participant-level phenotype and genotype data of Qatar Biobank/Qatar Genome Project in public databases. However, access to QBB/QGP data can be obtained through an established ISO-certified process by submitting a project request at https://researchportal.q phi.org.qa/login, which is subject to approval by the QPHI IRB committee.

**Acknowledgements.** We thank Ikhlak Ahmed, Mohammed ElAnbari, Najeeb Syed, Anjanarani N and Mashael Alshafai for their involvement in the preliminary analysis or management.

**Author contribution.** Conceptualization, P.V.J.; Methodology, D.V., K.B., A.I., M.A. and S.G.; Formal Analysis, D.V., K.B., A.I., A.A. and M.A.; Investigation, All; Data Curation, K.B.; Writing – Original Draft Preparation, P.V.J. and D.V.; Writing – Review & Editing, All; Supervision, M.W.A., M.P. and P.V.J. Author contributions for the QGP Research Consortium are provided in the supplementary material.

**Financial support.** We thank the QBB and QGP (now QPHI) for providing in-kind funding through access to the whole genome sequencing data and associated phenotypic data for this study as part of the QGP Research Consortium. PVJ received faculty research funding from the College of Health & Life Sciences, HBKU. The funders had no role in the interpretation of data.

**Competing interests.** M.P. currently receives partnership funding, paid to the University of Liverpool, for the following: MRC Clinical Pharmacology Training Scheme (co-funded by MRC and Roche, UCB, Eli Lilly and Novartis) and the MRC Medicines Development Fellowship Scheme (co-funded by MRC and GSK, AZ, Optum and Hammersmith Medicines Research). He has developed an HLA genotyping panel with MC Diagnostics but does not benefit financially from this. He is part of the IMI Consortium ARDAT (www.ardat.org); none of these funding sources have been used for the current research. MP is also Vice Chair of the Qatar Precision Health Initiative International Scientific Advisory Committee. The remaining authors have nothing to disclose.

**Ethics statement.** All participants provided written informed consent for the study and the study was approved by the QBB Institutional Review Board (https://www.qatarbiobank.org.qa/): QF-QGP-RES-PUB-008.

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
