## [Reviewer Report]

This study is an important contribution to the field of pharmacogenomics, offering valuable insights into the Qatari population. However, its clinical impact is limited by a lack of real-world applicability, incomplete methodology, and an overemphasis on frequency data without linking findings to patient outcomes.

The inclusion of 14,354 Qatari genomes ensures robust data and allows for detailed analysis of clinically actionable pharmacogenomic variants.

A few comments for improvement

The manuscript frequently implies that findings in Qatar can be extrapolated to the broader Middle Eastern population without sufficient justification or data. The assumption that diplotype-to-phenotype relationships are universally valid across populations lacks direct evidence. While existing literature supports this, population-specific epistatic interactions could alter metaboliser status.

Along the same line, the study assumes that identified diplotypes confer metaboliser effects identical to those observed in European populations. This overlooks potential differences due to genetic or environmental factors in Qataris, such as dietary or cultural influences on drug metabolism.

Furthermore, the study contrasts Qatari data with European and global populations but does not account for differences in healthcare practices or prescribing patterns, which may also influence pharmacogenomic actionability.

The manuscript discusses the “actionable” frequencies for different psychotropic drugs but does not adequately address how these frequencies translate into clinical practice (e.g., prescribing decisions or patient outcomes). More clarity on what constitutes “actionable” would strengthen the analysis. Discuss the magnitude of clinical benefits from implementing pharmacogenomic testing. For example, how much improvement in treatment response or reduction in adverse drug reactions is expected?

There is insufficient detail on how rare variants or structural variants impacting CYP enzymes were handled. These could affect metaboliser predictions and actionable frequencies.

The statistical methods lack depth in addressing multiple testing corrections for population comparisons, particularly for the large number of drugs and genotypes analysed.

Provide more detailed tables and figures linking diplotypes, metaboliser phenotypes, and actionable clinical recommendations. Incorporate confidence intervals and effect sizes in statistical comparisons.

Linking genotype data to actual clinical outcomes, such as adverse drug reactions or treatment efficacy, would significantly enhance the translational impact of the study.

Discuss real-world implementation scenarios, including potential challenges, and suggest pathways for integrating pharmacogenomics into routine care.

---

## [Reviewer Report]

The study addresses the pharmacogenomic landscape of antidepressants and antipsychotics in Qatar. This is highly relevant in advancing precision medicine for diverse populations. The analysis of 14,354 Qatari genomes provides robust data and significant insights into genetic variability and its implications for pharmacogenomics. The study relies on guidelines from CPIC and DPWG, ensuring clinical relevance and potential applicability of the findings. The inclusion of comparisons between Qatari and global populations (e.g., 1000 Genomes data) highlights regional differences in pharmacogenomic variability, underlining the need for population-specific guidelines.

Here are some points for the authors to consider to improve the manuscript.

The study focuses only on CYP2C19, CYP2D6, CYP2B6, and CYP3A4, which, while significant, do not cover all relevant pharmacogenomic markers for psychotropic medications. Why were only these genes selected? Incorporating additional genes could strengthen the analysis.

The authors acknowledge that phenotype predictions are based on literature from other populations. This approach may overlook population-specific variants, potentially limiting the accuracy and applicability of the findings.

The authors mention the limitation of excluding rare variants but do not propose solutions or future directions to address this gap.

Importantly, the study does not discuss how thresholds for clinical actionability were defined or whether they align with local clinical practices and prescribing patterns in Qatar. The authors should consult local stakeholders and add more information on prescribing patterns in Qatar and the practical clinical relevance and prescriber views on this information.

---

## [Reviewer Report]

The manuscript has been improved, but several major issues remain that need to be addressed.

The revised text in lines 251–263 does not adequately respond to the reviewer’s original comment regarding the need to justify the assumption that identified diplotypes confer metaboliser phenotypes identical to those in European populations. The potential influence of population-specific genetic and environmental factors in Qataris has not been sufficiently considered or discussed. Furthermore, the authors’ claim that a substantial proportion of Qataris require treatment modifications lacks strong and robust justification.

The conclusion should more clearly acknowledge the current limitations and evidence gaps, ensuring that any caveats are explicitly stated and that conclusions are appropriately cautious. Related updates should also be made to the abstract, impact statement, and the discussion’s concluding section to reflect this.

More detail is needed regarding the use of the Benjamini-Hochberg procedure—specifically, the false discovery rate (FDR) threshold employed and the rationale for its selection.

Given the inclusion of new analyses, the methods section now requires significant revision. It should include comprehensive descriptions of all methods used, data sources, and the rationale for key analytical decisions. This level of detail is essential for readers to assess the validity and reproducibility of the findings.

The new Supplementary Figure 1 should be described and contextualised within the methods, results, and discussion sections. Without this, it will be difficult for readers to interpret its significance. Additionally, the rationale for focusing on a subset of genes and not applying similar analyses to other genes with metaboliser phenotypes should be explained.

The utility of Figure 1A and Figure 2B is unclear—these appear to be redundant and their added value should be clarified or the figures removed.

Finally, the authors have misinterpreted the statement that diplotype-to-phenotype relationships are universally valid across populations. While this may be theoretically true, it does not hold in practice due to inter-population genetic variability and gene-environment interactions.